# Numerical Simulation of Sulfur Deposition in Wellbore of Sour-Gas Reservoir

Xiao Guo [1,*], Pengkun Wang [1], Jingjing Ma [2] and Changqing Jia [2]

1   State Key Laboratory of Oil and Gas Reservoir Geology and Exploitations, Southwest Petroleum University, Chengdu 610500, China
2   PetroChina Southwest Oil & Gas Field Company, Chengdu 610500, China
*   Correspondence: guoxiao@swpu.edu.cn

**Abstract:** Sulfur deposition has an important effect on the productivity of sour-gas wells. Accurately predicting the occurrence of sulfur deposition and the location and amount of sulfur deposition in wellbore can effectively guide the production of gas wells. In this paper, the wellbore sulfur deposition model, pressure model, and transient temperature model are established for various well types. Then, numerical simulations of sulfur deposition in the sour-gas well were conducted by coupling these models. Examples show that the proposed methodology has high accuracy, and the average relative error of the calculated results is 3.61%. Based on the model, a sensitivity analysis was performed on the factors affecting sulfur deposition. The results show that with the increase of wellbore inclination angle, the critical sulfur carrying velocity first increased and then decreased, and the maximum critical velocity is about 30% larger than that of the vertical section. The amount of wellbore sulfur deposition increases with increased production time and decreased wellbore pressure, and the amount of wellbore sulfur deposition decreases with increased gas production rate, $H_2S$ content, and inclination angle. The results suggest that the sour-gas reservoir should be developed with the horizontal or deviated well, timely adjust the production system, and keep the gas-well production higher than the critical flow rate as much as possible. At the same time, wellbore heating and insulation, pre-cleaning technology, and the closely implemented sulfur deposition prevention technology in the middle and late stage of development can be adopted to reduce the occurrence of sulfur deposition to ensure the safe and efficient development of high-sulfur gas wells.

**Keywords:** sour-gas well; sulfur deposition; critical sulfur carrying velocity; coupling model; numerical simulation

## 1. Introduction

With the increasing global demand for energy and the requirements of environmental protection, the proportion of natural gas in primary energy consumption is increasing year by year. Sour-gas reservoir development plays an important role in natural gas exploitation, and the sour-gas reservoirs are widely distributed around the world. Different from conventional gas reservoir development, the existence of elemental sulfur greatly affects the production of sour-gas reservoirs. With the decrease of temperature and pressure, elemental sulfur will precipitate from the natural gas when the sulfur concentration exceeds the sulfur solubility, thereby blocking the pore and reducing the permeability of the reservoir. At the same time, sulfur precipitates from the wellbore will be deposited on the wellbore [1], which can reduce the productivity of the gas wells and even lead to dead gas wells due to sulfur plugging. Serious harm from sulfur deposition to gas-well production has been reported in some high-sulfur content wells [2]. Adopting a reasonable production system and effective prevention measures of sulfur deposition can reduce the harm of sulfur deposition to gas-well production. Therefore, it is of great significance to ascertain whether elemental sulfur precipitation and deposition occur and accurately predict the location and amount of sulfur precipitated and deposition in wellbore for guiding the formulation and

application of prevention and control measures of sulfur deposition to ensure the normal production of sour-gas wells.

At present, many scholars have carried out relevant studies on sulfur deposition. Chrastil [3] first proposed a thermodynamic empirical model, which was widely used to calculate sulfur solubility. Through experimental fitting and theoretical research, many scholars have improved the Chrastil model to make it more accurate and more applicable [4–7]. In recent years, some scholars have used the machine-learning method [8] and the support vector machine method [9] to calculate sulfur solubility. Zhang [10] used the static method to measure the sulfur solubility in natural gas samples and conducted a comparative analysis on various solubility calculation models. The results showed that the fitted Chrastil model had good applicability.

Abou-Kassem and Jamal [11] analyzed the influence of fluid flow rate on sulfur deposition through a core flooding experiment and proposed the concept of critical sulfur carrying velocity. Hu [1] studied sulfur glomeration mechanism and established a calculation model of critical sulfur-carrying velocity suitable for vertical Wells. Al-Jaberi used WinProp from CMG for phase behavior of a generalized deep sour-gas well in the Middle East; the results showed that sulfur deposition occurs in the tubing [12]. Kuo and Closmann [13,14] established a one-dimensional radial flow model to study the effects of production rate, wellbore radius, and well spacing on the sulfur deposition. Mahmoud and Al-Majed [15,16] developed an analytical model to predict the formation damage due to sulfur deposition, and the results showed that sulfur deposition mainly occurs near the wellbore. Based on a conventional black-oil reservoir simulator and a dual porosity media analytical model, Hu [17,18] established a reservoir damage model in the presence of non-Darcy flow to analyze the relationship between sulfur deposition and saturation degree of irreducible water and the effect of sulfur deposition on well performance. Qin and Liu [19] introduced the concept of sulfur-deposition-equivalent wellbore radius (SDER) to develop a productivity equation for fractured gas wells that considers the effect of sulfur deposition. Yang [20] established a mathematical model that can be used for estimating productivity of sour-gas wells with the horizontal well type. Zou [21] developed a numerical model considering the damage of sulfur deposition with pressure change on reservoir porosity and permeability to predict the production from fractured horizontal wells in high-sulfur-content gas reservoirs and analyzed the influence of sulfur deposition on the production of fractured horizontal wells and the effects of hydraulic fracture parameters on production.

In terms of gas-well modeling, Dou [22] established a mathematical model for wellbore flow and heat transfer during the formation of high-sulfur gas invasion, which can be used to calculate wellbore temperature and pressure profile. However, the model did not consider the influence of sulfur deposition. Liu [23] established a temperature–pressure coupling model for high-pressure and high-temperature gas wells and without considering the influence of well type and sulfur deposition. Tan [24] studied the inflow characteristics of the horizontal gas well with sulfur deposition by experiment. Haq [25] modified the dynamic gas material balance equation and proposed a method to determine whether sulfur deposition occurred in the wellbore. Based on the volumetric source for horizontal wells of sulfur gas reservoirs, Shao [26] presented a semi analytical coupled reservoir/wellbore model, which can be used to predict the production and inflow profile along the horizontal well in sour-gas reservoirs. Liu [27] established a model for calculating bottomhole pressure suitable for sour-gas wells. This model has high accuracy in predicting bottomhole pressure, but it is only applicable to vertical wells and cannot calculate the amount of sulfur deposition in the wellbore.

In conclusion, the existing research is mainly aimed at the mathematical models of sulfur deposition and the productivity equations of the sour-gas well, but the numerical simulation methods that can be used to simulate the comprehensive situation of sulfur deposition in gas wells with different well types have not been formed. In this study, a prediction model of wellbore sulfur deposition considering different well trajectory types

was established. By coupling the prediction model with the wellbore pressure model and unsteady temperature model, the numerical simulation method of sulfur deposition in high-sulfur gas wells was obtained. The model was solved by numerical approximation and iterative methods, and it was verified by a typical high-sulfur gas well. The influence of different factors on sulfur deposition in gas wells was analyzed, and the corresponding prevention, control, and treatment measures of sulfur deposition were put forward. This method has the characteristics of high precision and wide application range and can effectively and accurately predict the sulfur precipitation and deposition in wellbore. The results of this paper are helpful to guide the efficient and safe development of sour-gas wells.

## 2. Wellbore Sulfur Deposition Prediction Model

### 2.1. Model Assumption

The following assumptions were made for this study:

(a) When the gas production flow velocity in the wellbore is greater than the critical sulfur carrying velocity, the precipitated sulfur does not deposit.

(b) When the gas production flow velocity is less than the critical sulfur-carrying velocity, the precipitated liquid sulfur will fall to the bottom of the well, and the precipitated solid sulfur will adhere to the wellbore to form sulfur scale deposition.

(c) The influence of differential pressure force, additional mass force, Bassett force, and Magnus force is ignored.

(d) The liquid sulfur droplets are ellipsoid and solid sulfur particles are spheroids in the wellbore, and their morphology does not change during movement.

(e) The flow in wellbore is one-dimensional, stable, and linear.

### 2.2. Sulfur Solubility

According to the thermodynamic empirical model proposed by J. Chrastil [3], the formula for calculating sulfur solubility is:

$$C = \rho_g{}^k \exp\left(\frac{A}{T} + B\right) \tag{1}$$

Based on the experimental data, Roberts [4] fitted the Chrastil model and obtained empirical correlation with constant coefficients:

$$C = \rho_g{}^4 \exp\left(-\frac{4666}{T} - 4.5711\right) \tag{2}$$

With the decrease of temperature and pressure, sulfur solubility will decrease. When the instantaneous sulfur solubility, $C_s$, is less than the initial sulfur solubility, $C_o$, the sulfur will be precipitated from the gas. The formula for calculating the amount of sulfur precipitation is:

$$Q_s = (C_o - C_s)Q_g \tag{3}$$

According to the results of the previous studies on the phase state of sulfur, this paper considers that when the temperature is greater than 393 K, the precipitated sulfur is liquid, and when the temperature is less than 393 K, the precipitated sulfur is solid.

### 2.3. Critical Liquid Sulfur Carrying Velocity

After the liquid sulfur is precipitated, it is mainly affected by gravity, $G$, buoyancy, $F_f$, and drag, $F_D$, as well as friction, f, and support force, $F_N$, of the inner wall of the wellbore, as shown in Figure 1.

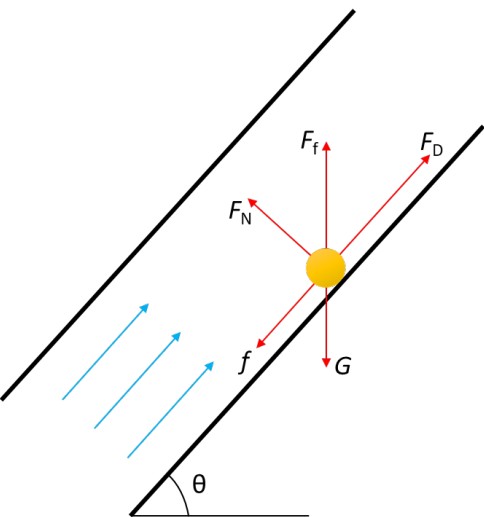

**Figure 1.** Stress analysis of liquid sulfur droplet in wellbore.

The expression of each force can be found in literature [1] and Equations (4) and (5).

$$F_N = (G - F_f) \cos\theta \tag{4}$$

$$f = \lambda F_N \tag{5}$$

When the resultant force on the liquid sulfur is 0, the following equation can be obtained:

$$F_D - (G - F_f) \sin\theta - f = 0 \tag{6}$$

Substituting Equations (4) and (5) into Equation (6):

$$v_s = v_m - \sqrt{\frac{2(\rho_s - \rho_m)(\sin\theta + \lambda\cos\theta)V_s g}{C_D S \rho_m}} \tag{7}$$

Then the formula for calculating critical carrying velocity of liquid sulfur can be obtained:

$$v_{gcr} = \sqrt{\frac{2(\rho_s - \rho_m)(\sin\theta + \lambda\cos\theta)V_s g}{C_D S \rho_m}} \tag{8}$$

According to Li's model [28], liquid sulfur droplets in wellbore are ellipsoid, and the volume calculation formula is:

$$V_s = \frac{4}{3} S h_s \tag{9}$$

The critical carrying velocity should be the gas production flow velocity at which the largest diameter of the droplet in the wellbore can still be taken out of the wellhead. Therefore, the maximum diameter of the droplet is the droplet height, and the droplet height can be calculated by:

$$h_s = d_{s,\,max} = \frac{\sigma N_{we}}{\rho_m v_{gcr}^2} \tag{10}$$

Substituting Equations (9) and (10) into Equation (8):

$$v_{gcr} = \sqrt[4]{\frac{8(\rho_s - \rho_m)(\sin\theta + \lambda\cos\theta)\sigma N_{we} g}{3 C_D \rho_m^2}} \tag{11}$$

The critical Weber number of droplets in the gas flow is 20~30, and the effective inflow area of ellipsoid droplets is close to 100%. Therefore, take $N_{we} = 30$, $C_D = 1$ and substitute into Equation (11):

$$v_{gcr} = 2.99\sqrt[4]{\frac{(\rho_s - \rho_m)(\sin\theta + \lambda\cos\theta)\sigma g}{\rho_m{}^2}} \tag{12}$$

*2.4. Critical Solid Sulfur Carrying Velocity*

According to the experimental study on the cutting behavior in the wellbore, solid sulfur particles will form rolling motions when the wellbore inclination angle $\alpha$ is high, as shown in Figure 2.

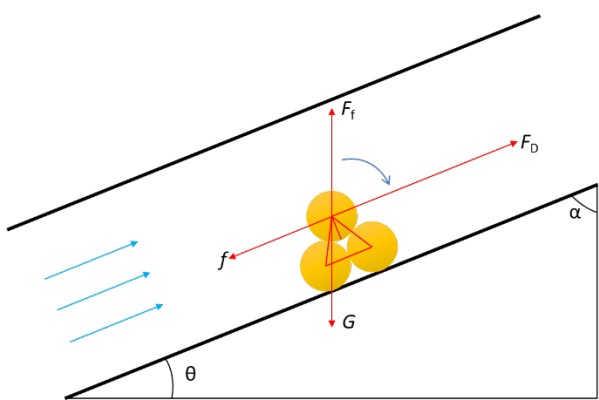

**Figure 2.** Stress analysis of solid sulfur particle in wellbore.

Establish the torque equilibrium equation:

$$(F_D - f)\frac{d_s}{2}\cos\theta - (G - F_f)\frac{d_s}{2}\sin\left(\frac{\pi}{3} + \theta\right) = 0 \tag{13}$$

Substituting Equations (4) and (5) into Equation (13):

$$v_{gcr} = \sqrt{\frac{2(\rho_s - \rho_m)\left[\sin\left(\frac{\pi}{3} + \theta\right) + \lambda\cos^2\theta\right]V_s g}{S\rho_m C_D\cos\theta}} \tag{14}$$

The volume calculation formula of solid sulfur particles is:

$$V_s = \frac{1}{6}\pi d_s{}^3 \tag{15}$$

The stressed area of drag force is:

$$S = \frac{1}{4}\pi d_s{}^2 - \frac{1}{8}d_s{}^2\left(\frac{\pi}{3} - \sin\frac{\pi}{3}\right) = 0.7628d_s{}^2 \tag{16}$$

Substituting Equations (15) and (16) into Equation (14):

$$v_{gcr} = \sqrt{\frac{\pi(\rho_s - \rho_m)\left[\sin\left(\frac{\pi}{3} + \theta\right) + \lambda\cos^2\theta\right]d_s g}{2.2884\rho_m C_D\cos\theta}} \tag{17}$$

When the inclination angle is low, solid sulfur particles are suspended in gas and move towards the wellhead. Thus, the influence of solid sulfur particles on wellbore friction and support force can be neglected, and the torque balance equation is:

$$F_D\sin\theta + F_f - G = 0 \tag{18}$$

Then the critical carrying velocity is:

$$v_{\text{gcr}} = \sqrt{\frac{2(\rho_s - \rho_m)V_s g}{S\rho_m C_D \sin\theta}} \tag{19}$$

Substituting Equations (15) and (16) into Equation (19):

$$v_{\text{gcr}} = \sqrt{\frac{\pi(\rho_s - \rho_m)d_s g}{2.2884\rho_m C_D \sin\theta}} \tag{20}$$

According to the experimental results of Ford [29], the critical suspension velocity is roughly the same as the critical roll velocity when the inclination angle is 40~60°. Considering the continuity of the relationship between the solid-sulfur-carrying velocity and the inclination angle, the paper deems that when the wellbore inclination angle is less than 45°, the solid sulfur particles mainly move to the wellhead by suspension, and when the inclination angle is greater than 45°, the solid sulfur particles mainly move to the wellhead by rolling. Therefore, the calculation formula of critical solid-sulfur-carrying velocity is:

$$v_{\text{gcr}} = \begin{cases} \sqrt{\dfrac{\pi(\rho_s - \rho_m)d_s g}{2.2884\rho_m C_D \sin\theta}} & \theta > 45° \\[3mm] \sqrt{\dfrac{\pi(\rho_s - \rho_m)\left[\sin\left(\frac{\pi}{3}+\theta\right)+\lambda\cos^2\theta\right]d_s g}{2.2884\rho_m C_D \cos\theta}} & \theta \le 45° \end{cases} \tag{21}$$

Within this equation, the drag coefficient, $C_D$, adopts the multi-gene GP model established by Barati [30]:

$$\begin{aligned} C_D = \quad & 8\times10^{-6}\left[\left(\frac{Re}{6530}\right)^2 + \tanh(Re) - \frac{8\ln(Re)}{\ln(10)}\right] - 0.4119e^{-\frac{2.08\times10^{43}}{(Re+Re^2)^4}} \\ & -2.1344e^{-\frac{\left[\ln\left(Re^2+10.7563\right)/\ln(10)\right]^2+9.9867}{Re}} + 0.1357e^{-\frac{(Re/1620)^2+10370}{Re}} \\ & -8.5\times10^{-3}\frac{2\ln\{\tanh[\tanh(Re)]\}/\ln(10)-2825.7162}{Re} + 2.4795 \end{aligned} \tag{22}$$

$Re$ is the Reynolds number, and its expression is:

$$Re = \frac{\rho_m v_m D}{\mu_m} \tag{23}$$

### 2.5. Diffusion Deposition Model of Sulfur

When solid sulfur is precipitated and the gas flow velocity in wellbore is less than the critical sulfur-carrying velocity, sulfur will migrate to the pipe wall and then attach and deposit on the pipe wall. The mass transfer flux is expressed as:

$$J_s = \begin{cases} 0 & x_s \le x_s{}^{jb} \\ k_s\rho_m M_s\left(x_s - x_s{}^{jb}\right) & x_s > x_s{}^{jb} \end{cases} \tag{24}$$

Within this equation:

$$k_s = j_D v_m$$

$$j_D = \begin{cases} 0.023(Re)^{-0.17}(Sc)^{-\frac{2}{3}} & Re = 4000 \sim 60000, Sc = 0.6 \sim 3000 \\ 0.0149(Re)^{-0.12}(Sc)^{-\frac{2}{3}} & Re = 10000 \sim 400000, Sc > 10 \end{cases}$$

$$Sc = \frac{\mu_m}{\rho_m D_s{}^m}$$

With the increase of sulfur deposition, the sulfur scale thickness increases and the wellbore diameter decreases, and its change equation is:

$$\frac{\mathrm{d}D}{\mathrm{d}t} = -\frac{J_s}{\rho_s} \tag{25}$$

## 3. Wellbore Pressure Model and Transient Temperature Model

### 3.1. Model Assumption

The following assumptions were made:

(a) The fluid in the wellbore is a one-dimensional steady flow.
(b) The change of physical property parameters, such as density during fluid flow, is considered.
(c) Changes in wellbore diameter (except those due to sulfur deposition) are not considered, i.e., energy loss due to wellbore diameter change is not considered.
(d) Formation temperature remains constant.
(e) The effect of sulfur deposition on temperature distribution is considered in the wellbore transient temperature model.
(f) The initial value of wellbore temperature distribution is given by the original formation temperature and geothermal gradient.

### 3.2. Wellbore Pressure Model

Divide the wellbore into $N$ sections of equal depth ($i = 1, 2, 3, \ldots , N$). The pressure drop of the section is shown in Figure 3.

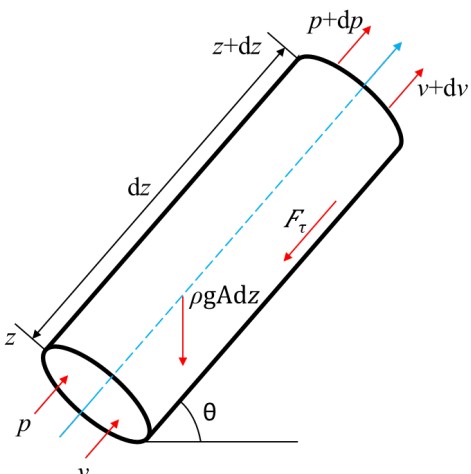

**Figure 3.** Schematic diagram of section pressure drop.

According to the law of conservation of mass:

$$\frac{\mathrm{d}(\rho v A)}{\mathrm{d}z} = 0 \tag{26}$$

The external force on the section is equal to the change in momentum of the fluid in the section:

$$\sum F_z = \rho A \mathrm{d}z \frac{\mathrm{d}v}{\mathrm{d}t} \tag{27}$$

The external forces on the section mainly include the gravity, $F_G$, friction, $F\tau$, and the external forces generated by the pressure difference. The expressions of each force are:

$$F_G = -\rho_m g A \mathrm{d}z \sin\theta \tag{28}$$

$$F_\tau = -\lambda \frac{\rho_m v_m^2 A}{2D} dz \tag{29}$$

$$pA - (p + dp)A = -Adp \tag{30}$$

Combining Equations (26)–(30), the pressure drop of wellbore section is:

$$\frac{dp}{dz} = -\rho_m g \sin\theta - \lambda \frac{\rho_m v_m^2}{2D} - \rho_m v_m \frac{dv}{dz} \tag{31}$$

Within this equation:

$$\rho_m = \rho_g \left(1 - \frac{Q_s}{Q_s + Q_g}\right) + \rho_s \frac{Q_s}{Q_s + Q_g}$$

### 3.3. Wellbore Transient Temperature Model

The heat transfer of the section is shown in Figure 4.

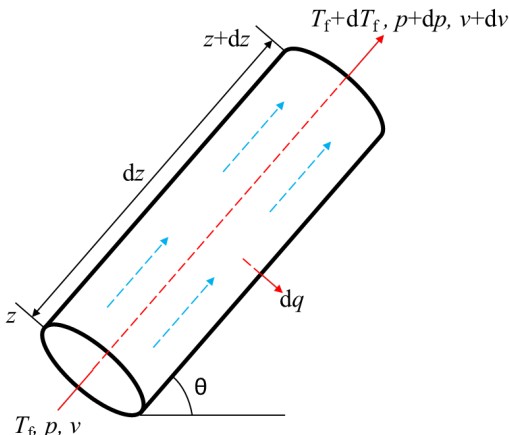

**Figure 4.** Schematic diagram of section heat transfer.

According to the law of conservation of energy:

$$\frac{dh}{dz} = \frac{1}{W}\frac{dq}{dz} - v_m \frac{dv}{dz} - g\sin\theta + \frac{\lambda v_m^2}{2ddz} \tag{32}$$

According to the basic theory of thermodynamics:

$$dh = \left(\frac{\partial h}{\partial T}\right)_p dT + \left(\frac{\partial h}{\partial p}\right)_T dp \tag{33}$$

According to the definition of specific heat of mixed fluid and Joule Thomson coefficient:

$$c_{pm} = \left(\frac{\partial h}{\partial T}\right)_p \tag{34}$$

$$\alpha_H = \left(\frac{\partial T}{\partial p}\right)_h = -\frac{(\partial h/\partial p)_T}{(\partial h/\partial T)_p} = -\frac{1}{c_{pm}}\left(\frac{\partial h}{\partial p}\right)_T \tag{35}$$

Substituting Equations (33)–(35) into Equation (32):

$$\frac{dT}{dz} = \frac{1}{c_{pm}}\left(\frac{1}{W}\frac{dq}{dz} - v_m\frac{dv}{dz} - g\sin\theta + \frac{\lambda v_m^2}{2ddz}\right) + \alpha_H \frac{dp}{dz} \tag{36}$$

According to the transient heat transfer differential equation of wellbore and formation, the thermal conductivity differential equation of wellbore and formation is:

$$\frac{\partial^2 T_1(r,t)}{\partial r^2} + \frac{1}{r}\frac{\partial T_1(r,t)}{\partial r} = \frac{1}{\alpha_1}\frac{\partial T_1(r,t)}{\partial t} \tag{37}$$

Within this equation:

$$\alpha_1 = U_{to}(\rho c)_h{}^{-1}$$

According to the Fourier's theorem:

$$\frac{dq}{dz} = -2\pi r_h U_{to}\frac{\partial T_1}{\partial r}\bigg|_{r = r_h{}^+} \tag{38}$$

Considering the initial conditions, the temperature at any location of the wellbore is the initial formation temperature, and the formation temperature at infinity is the initial formation temperature. After dimensionless processing, the following results can be obtained:

$$T_D = -\frac{2\pi U_{to}}{W dq/dz}(T_1 - T_{ei}) \tag{39}$$

After Laplace transform, Bessel equation solution, and Stefest numerical inversion, Equation (40) can be obtained as:

$$\frac{dq}{dz} = -\frac{2\pi U_{to}}{W T_D}(T_1 - T_{ei}) \tag{40}$$

The dimensionless time function is [31,32]:

$$T_D = \begin{cases} 1.1281\sqrt{t_D}\left(1 - 0.3\sqrt{t_D}\right) & t_D \leq 1.5 \\ (0.5\ln t_D + 0.4063)\left(1 + \frac{0.6}{t_D}\right) & t_D > 1.5 \end{cases} \tag{41}$$

Within this equation:

$$t_D = 7.5 \times 10^7 \frac{t}{r_h{}^2}$$

Substituting Equation (40) into Equation (36):

$$\frac{dT}{dz} = \frac{1}{c_{pm}}\left(-\frac{2\pi U_{to}}{W T_D}(T_1 - T_{ei}) - v_m\frac{dv}{dz} - g\sin\theta + \frac{\lambda v_m{}^2}{2ddz}\right) + \alpha_H\frac{dp}{dz} \tag{42}$$

Within this equation, considering the influence of sulfur deposition, the total wellbore heat transfer coefficient is:

$$U_{to} = \left(R_j + \frac{r_{to}\ln\frac{r_{ti}}{r_{ti}-\Delta r}}{\lambda_s}\right)^{-1}$$

Consider Equation (42) as an ordinary differential equation:

$$\begin{aligned} T_{f,out} = & \ T_{ei,out} + \exp\left(\frac{\Delta z}{A}\right)(T_{f,in} - T_{ei,in}) \\ & + A\left[1 - \exp\left(\frac{\Delta z}{A}\right)\right]\left(-\frac{g\sin\theta}{c_{pm}} + \phi + g_T\sin\theta + \frac{\lambda v_m{}^2}{2Dc_{pm}}\right) \end{aligned} \tag{43}$$

Within this equation:

$$A = \frac{WT_D c_{pm}}{2\pi U_{to}}$$

$$\phi = \alpha_H \frac{dp}{dz} - \frac{v_m}{c_{pm}} \frac{dv}{dz}$$

$$T_{ei} = T_i - g_T z \sin\theta$$

The initial value of wellbore temperature distribution is equal to the formation temperature:

$$T_{ini} = T_{ei} = T_i - g_T z \sin\theta \tag{44}$$

Given the bottom hole temperature condition, Equation (43) can be used to calculate the inlet and outlet temperature of each section.

## 4. Numerical Simulation of Sulfur Deposition in Sour-Gas Well

### 4.1. Model Solution

It is shown that some parameters interacted with each other in the wellbore pressure model and unsteady temperature model. Considering the influence of sulfur precipitation and deposition, the fluid physical properties and wellbore inner diameter will also change with time. In this paper, the sulfur deposition prediction model is coupled with the wellbore pressure and temperature model through numerical approximation, and the numerical simulation of sulfur deposition in the sour-gas well is carried out. The coupling solution steps are as follows:

(a)　Divide the wellbore into several sections according to the wellbore trajectory, and input data, such as section length and deviation angle for each unit.

(b)　Input initial physical parameters.

(c)　Calculate the initial value of wellbore temperature distribution, $T_{ini}$, by Equation (44). The initial value of wellbore pressure distribution, $p_{ini}$, can be obtained linearly from the initial bottomhole pressure to wellhead pressure.

(d)　Calculate the pressure of each section from the wellhead as the inlet end to obtain the wellbore pressure distribution, $p_{cal}$. The specific calculation process is:

　　a　Calculate the sulfur solubility of the section i by Equation (2) to determine whether the sulfur is precipitated. If so ($C_s > C_o$), determine whether the sulfur is deposited on the pipe wall according to the temperature conditions and gas flow velocity. If so (T > 119 °C and $v_{gcr} > v_m$), calculate the thickness of sulfur deposition and the wellbore diameter of the section *i* by Equations (24) and (25);

　　b　Calculate the fluid physical properties in section *i* considering sulfur precipitation;

　　c　Substitute the fluid physical properties into Equation (31) to calculate the initial value of the outlet pressure, $p_{i,out}$, in section *i*;

　　d　Since the initial pressure value cannot meet the accuracy requirements, it cannot be determined whether its value is larger or smaller than the actual value. Based on the idea of numerical approximation, assign the average value of $p_{i,in}$ and $p_{i,out}$ to $p_{i,in}$. Repeat steps a~c to get the revised value of the outlet pressure, $p_{i,out'}$;

　　e　Compare the initial value, $p_{i,out}$, and the revised value, $p_{i,out'}$. If the error is greater than the allowable error, $\delta$, assign $p_{i,out'}$ to $p_{i,out}$ and repeat steps a~e;

　　f　If the error is smaller than $\delta$, assign $p_{i,out}$ to $p_{i+1,in}$ and repeat steps a~f to get the wellbore pressure distribution, $p_{cal}$.

(e)　Calculate the temperature of each section from the bottom of the well as the inlet end to obtain the wellbore temperature distribution, $T_{cal}$. The specific calculation process is:

　　a.　Calculate thermal physical parameters of section *i*;

　　b.　Substitute thermal physical parameters and pressure calculated by step (d) into Equation (43) to calculate the outlet temperature, $T_{i,out}$, in section *i*;

      c.        Assign $T_{i,out}$ to $T_{i+1}$. Repeat steps a~c to get the wellbore temperature distribution, $T_{cal}$.

(f)    Compare the initial values, $T_{ini}$ and $p_{ini}$, with the calculated values, $T_{cal}$ and $p_{cal}$. If the error is greater than the allowable error, $\delta$, assign $T_{cal}$ and $p_{cal}$ to $T_{ini}$ and $p_{ini}$, and repeat steps (d)~(f) until the required accuracy is reached.

(g)    Assign $T_{cal}$ and $p_{cal}$ on day $t$ to $T_{ini}$ and $p_{ini}$ on day $t + 1$, and repeat steps (d)~(g) to get wellbore pressure and temperature distribution in different production times.

The specific coupling solution process is shown in Figure 5.

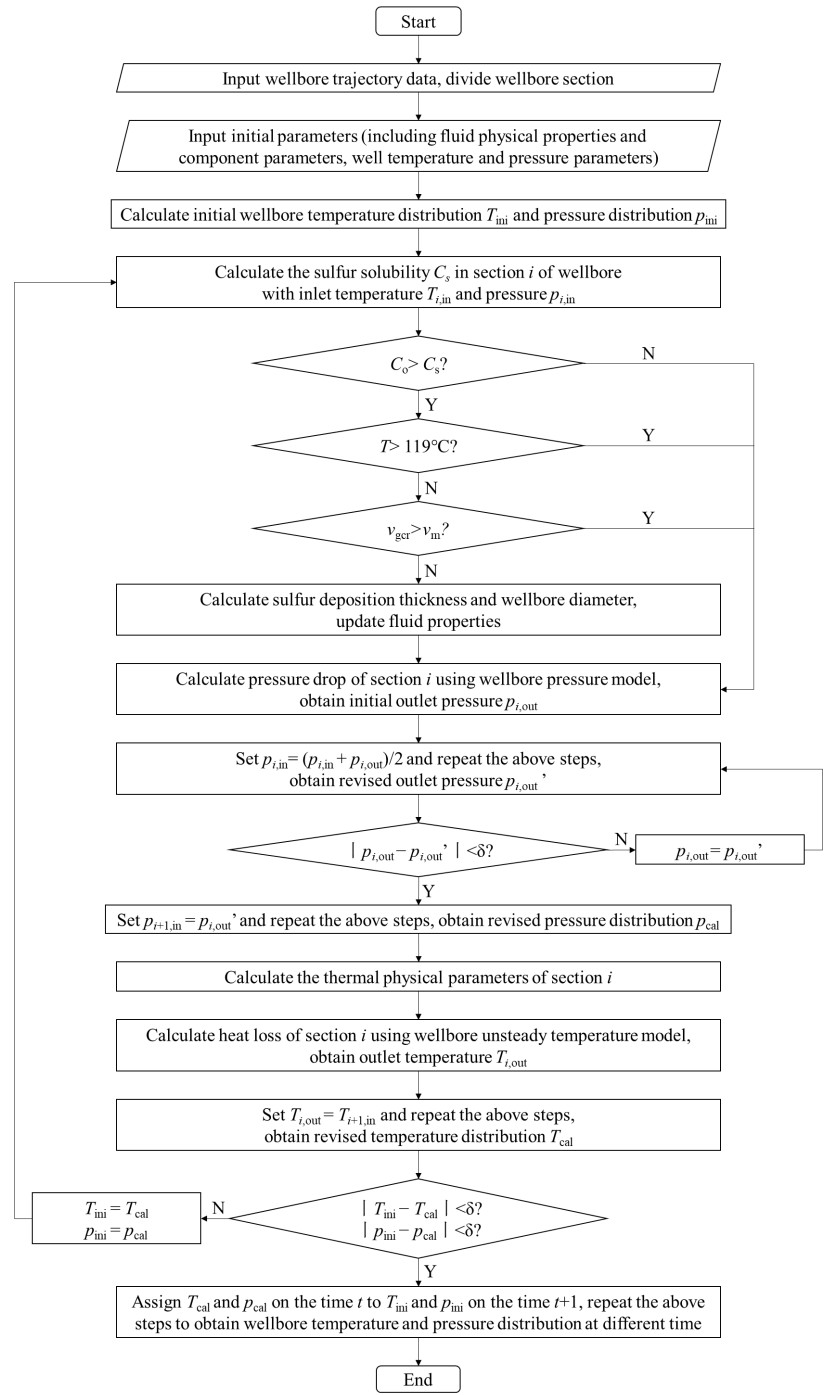

**Figure 5.** Flow chart of coupling solution.

### 4.2. Model Validation

For model validation, a typical high-sulfur content gas well (vertical well, located in Canada) was studied. The basic gas-well parameters and produced natural-gas parameters are shown in Table 1.

**Table 1.** Gas-well and produced natural-gas parameters.

| Well Parameters | | Natural Gas Main Components (%) | |
|---|---|---|---|
| Wellhead pressure (MPa) | 30.33 | $CH_4$ | 83.0 |
| Bottomhole temperature (°C) | 122.2 | $C_2H_6$ | 1.5 |
| Geothermal gradient (°C/m) | 0.024 | $C_{3+}$ | 0.5 |
| Gas production ($10^4$ m$^3$/d) | 1.62 | $H_2S$ | 10.4 |
| Well depth (m) | 4275.4 | $CO_2$ | 4.6 |

The calculation results of wellhead temperature, bottomhole pressure, and sulfur deposition location are shown in Table 2.

**Table 2.** Calculation results of model validation.

| Parameters | Calculated Value | Measured Value | Relative Error (%) |
|---|---|---|---|
| Wellhead temperature (°C) | 21.6 | 22.4 | 3.46 |
| Bottomhole pressure (MPa) | 40.4 | 41.2 | 2.01 |
| Sulfur deposition location (m) | 3440.0 | 3468.2 | 0.81 |

The calculated value is close to the measured value, and the relative errors are all less than 5%, indicating that the model in this paper has high accuracy.

The critical carrying velocity model in this paper was used to calculate the critical sulfur-carrying velocity at different well angles, and the basic parameters are shown in Table 3. It can be seen from Figure 6 that the calculation results and the experimental results of Ford [29] have the same change law. Because fluids and particle sizes used in the experiments and calculation model are different, there are some differences in the specific shape of these curves and values.

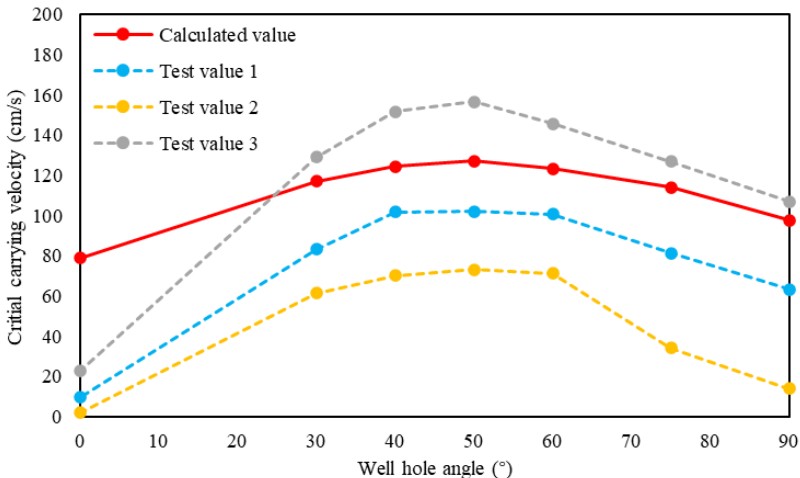

**Figure 6.** Relationship between critical sulfur-carrying velocity and inclination angle.

**Table 3.** Basic parameters used in critical carrying velocity model.

| Parameters | Value |
|---|---|
| Sulfur density (kg/m$^3$) | 2070 |
| Gas density (kg/m$^3$) | 130 |
| Friction coefficient | 0.1 |
| Diameter of sulfur particle (m) | $3.2 \times 10^{-3}$ |

The analysis of the relationship between critical sulfur-carrying velocity and wellbore inclination angle shows that the critical sulfur-carrying velocity first increases and then decreases with the increase of inclination angle, and the maximum critical sulfur-carrying velocity is about 30% lager than that in the vertical well section. Additionally, the critical sulfur-carrying velocity of the horizontal section is also slightly larger than that of the vertical section. Therefore, compared with the vertical wells, the gas production rate of the horizontal and deviated wells needs to be appropriately increased to ensure that the gas production flow velocity in the wellbore exceeds the critical sulfur-carrying velocity to avoid sulfur deposition.

## 5. Results and Discussion

### 5.1. Model Application

The sulfur depositions in four sour-gas wells located in a sour-gas reservoir in Sichuan Basin, China are simulated using the proposed methodology in this section. The basic gas-well parameters are shown in Table 4. The produced natural-gas parameters are shown in Table 5. And the calculation results are shown in Table 6.

**Table 4.** Basic gas-well parameters.

| Well Parameters | Well A | Well B | Well C | Well D |
|---|---|---|---|---|
| Well type | Vertical well | Horizontal well | Deviated well | Deviated well |
| Wellhead pressure (MPa) | 17.34 | 25.6 | 16.66 | 22.77 |
| Bottomhole temperature (°C) | 143.81 | 154.55 | 148.72 | 152.29 |
| Geothermal gradient (°C/m) | 0.021 | 0.021 | 0.021 | 0.021 |
| Gas production ($10^4$ m$^3$/d) | 24.2 | 8.0 | 64.0 | 28.6 |
| Well depth (m) | 6740 | 7180 | 6805 | 6963 |
| Kickoff point (m) | / | 6080 | 5920 | 6375 |
| Inclination angle (°) | 0 | 90 | 45 | 37 |

**Table 5.** Produced natural gas parameters.

| Components (%) | Well A | Well B | Well C | Well D |
|---|---|---|---|---|
| $CH_4$ | 88.817 | 90.391 | 90.554 | 87.774 |
| $C_2H_6$ | 0.036 | 0.031 | 0.027 | 0.033 |
| $C_{3+}$ | 0.004 | 0.006 | 0.009 | 0.003 |
| $CO_2$ | 5.093 | 4.152 | 3.614 | 5.247 |
| $N_2$ | 0.224 | 0.448 | 0.427 | 0.338 |
| $H_2S$ | 5.826 | 4.946 | 5.336 | 6.592 |
| $H_2$ | 0 | 0.010 | 0.001 | 0 |
| He | 0 | 0.016 | 0.032 | 0.013 |
| Total | 100 | 100 | 100 | 100 |

**Table 6.** Calculation results of model application.

| Parameters | Well A | Well B | Well C | Well D |
|---|---|---|---|---|
| Calculated bottomhole pressure (MPa) | 25.96 | 36.95 | 25.12 | 34.11 |
| Measured bottomhole pressure (MPa) | 25.12 | 34.39 | 24.88 | 33.21 |
| Relative error (%) | 3.25 | 6.93 | 0.97 | 2.63 |
| Calculated wellhead temperature (°C) | 41.38 | 23.87 | 73.27 | 52.93 |
| Measured wellhead temperature (°C) | 41.50 | 25.60 | 73.90 | 56.50 |
| Relative error (%) | 0.28 | 7.25 | 0.87 | 6.75 |

The calculation accuracy of the proposed model is high. The maximum relative error of calculating wellhead temperature and bottomhole pressure is 7.25%, and the average error is 3.61%. The proposed model is used to predict sulfur precipitation and deposition in the above four wells, and the results are shown in Figure 7.

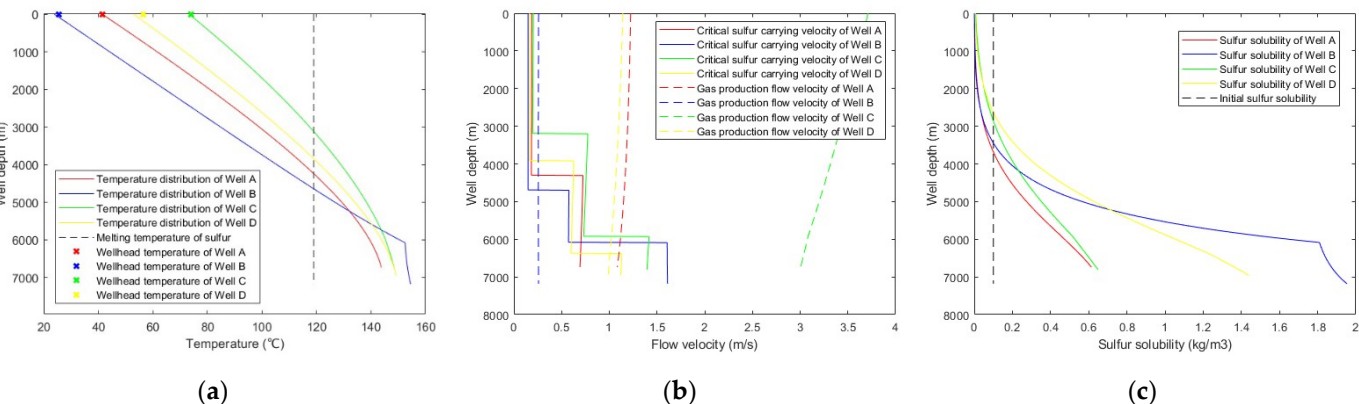

(**a**)　　　　　　　　　　(**b**)　　　　　　　　　　(**c**)

**Figure 7.** Predicted results with: (**a**) wellbore temperature distribution; (**b**) critical sulfur-carrying velocity versus gas production flow velocity; (**c**) sulfur solubility distribution along the wellbore.

As can be seen from Figure 7, the gas production flow velocities in Well A and Well C were greater than the critical sulfur-carrying velocities; thus, no sulfur depositions occur in Well A and Well C. The gas production flow velocities in Well B and Well D were lower than the critical sulfur-carrying velocities when the well depths exceed 4700 m and 6330 m, respectively. However, the sulfur solubility of these sections was higher than the initial sulfur solubility, and there is no sulfur precipitated; therefore, there is no sulfur deposition occurring in Well B and Well D either. However, sulfur precipitated in the middle and upper parts of the wellbore of each well, and sulfur may be deposited during the gathering and transportation process after the wellhead. It is necessary to take measures, such as sulfur solvent mixed injection and regular pigging, to prevent sulfur deposition from affecting production in the gathering and transportation process.

### 5.2. Sensitivity Analysis

5.2.1. Production Time

Assuming the production system remains unchanged, with the increase of production time, the sulfur solubility is basically unchanged, while the sulfur scale thickness increases monotonically. It is shown in Figure 8 that the sulfur scale thickness dose not decrease with the increase of well depth but increases first and then decreases. The reason is that within a certain pressure interval, gas viscosity does not monotonously increase or decrease with temperature but decreases first and then increases (as shown in Figure 9), which leads to changes in fluid flow parameters, thus resulting in a maximum sulfur precipitation rate and sulfur scale thickness. When the sulfur scale thickness is too large, sulfur plugging will form, and the productivity of the gas well will be seriously affected. Therefore, the production system should be dynamically adjusted to change the wellbore temperature and

pressure distribution so that the sulfur scale can be evenly distributed along the wellbore. Alternatively, the proposed model can be adopted to predict the sulfur deposition location and carry out regular preventive cleaning in the production process to effectively avoid the occurrence of sulfur plugging in the wellbore.

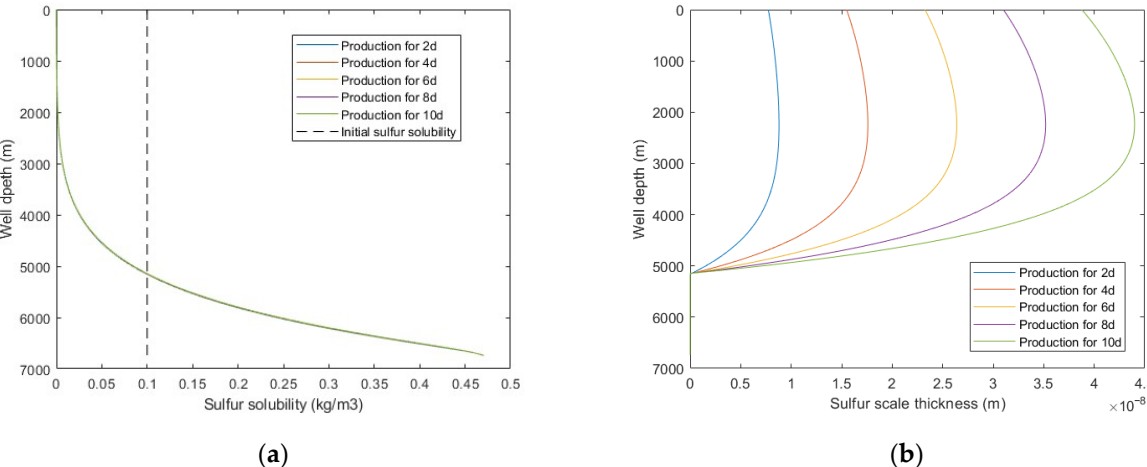

**Figure 8.** Calculated results with: (**a**) sulfur solubility distribution at different production times; (**b**) sulfur scale thickness distribution at different production times.

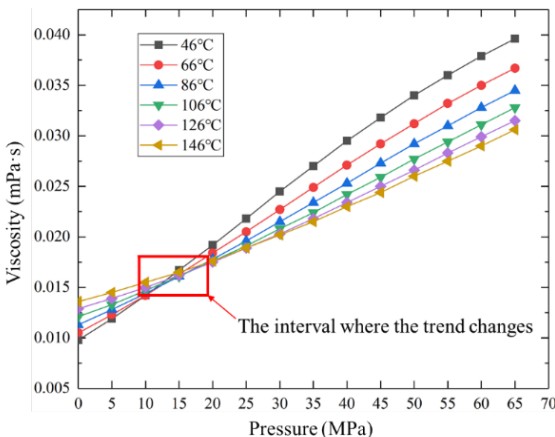

**Figure 9.** Gas viscosity under different temperatures and pressures.

### 5.2.2. Gas Production Rate

Figure 10 shows that under the same production time, increasing the gas production rate led to a larger sulfur solubility and sulfur scale thickness, and the sulfur deposition location moves closer to the wellhead. However, under the same gas production quantity, the larger the production rate, the smaller the sulfur scale thickness. In addition, when the gas production flow velocity exceeds the critical sulfur-carrying velocity, sulfur deposition is no longer generated in the wellbore. This is because the increased gas production rate increases the temperature, which increases the sulfur solubility and sulfur-carrying capacity of the produced natural gas. Therefore, the gas production rate should be greater than the critical sulfur-carrying velocity as far as possible to reduce sulfur deposition in the wellbore.

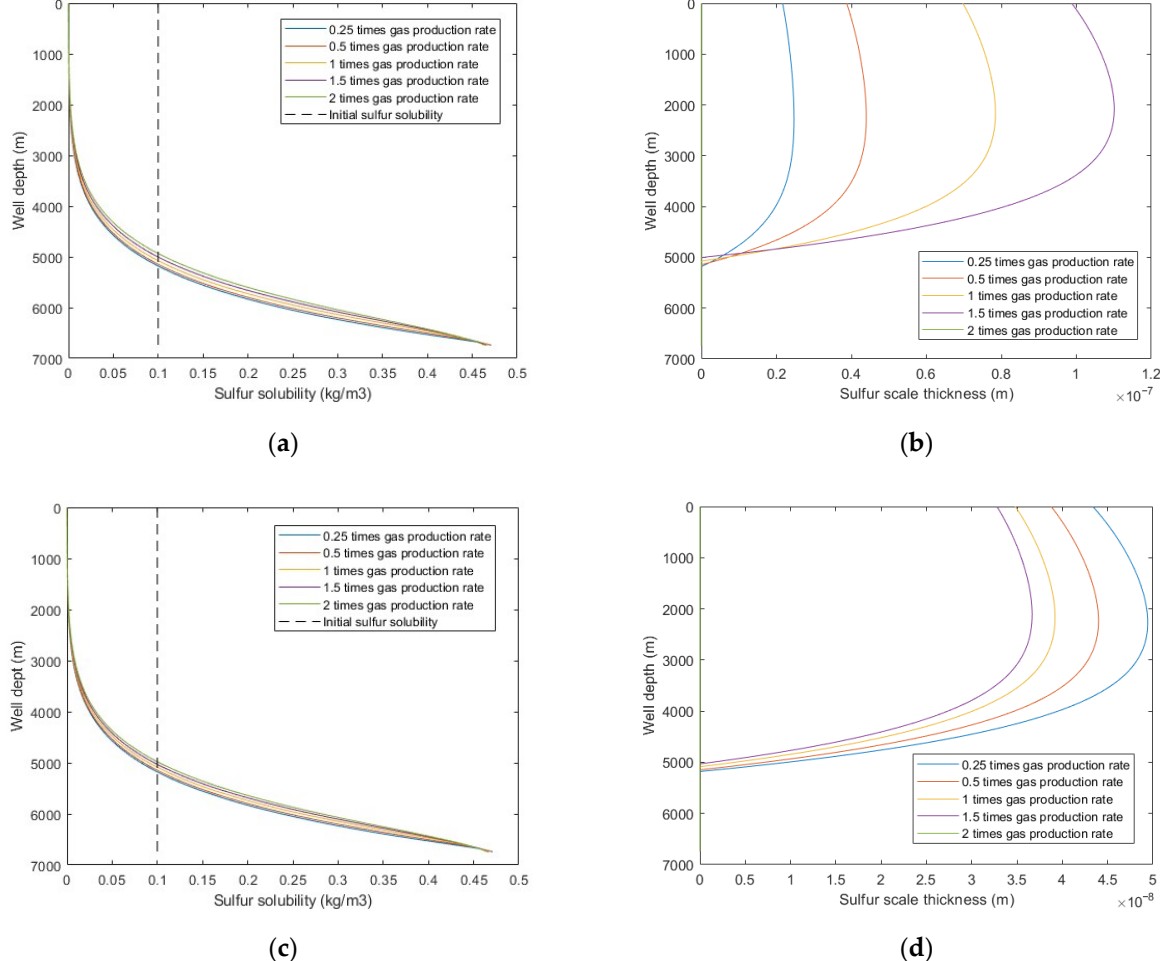

**Figure 10.** Calculated results with: (**a**) sulfur solubility distribution at different gas production rates (at same production time); (**b**) sulfur scale thickness distribution at different gas production rates (at same production time); (**c**) sulfur solubility distribution at different gas production rates (at same gas production quantity); (**d**) sulfur scale thickness distribution at different gas production rates (at same gas production quantity).

### 5.2.3. Wellbore Pressure

From Figure 11, it was observed that with decreasing wellbore pressure, the sulfur solubility decreased, sulfur scale thickness increased, and the sulfur deposition location moves closer to the bottomhole. When the wellbore pressure decreased to less than 50% of the original wellbore pressure, the sulfur deposition rate increases exponentially. This means that with the development of the sour-gas well, the occurrence of wellbore sulfur deposition will gradually extend from the wellhead to the bottomhole, the occurrence of sulfur deposition will become more serious, and the probability of wellbore sulfur plugging will be higher. Therefore, it is suggested to closely implement wellbore pre-cleaning and other technical measures to prevent sulfur deposition and to pay attention to observing and analyzing the production dynamic data in the middle and late stages of gas well production. When sulfur plugging occurs, timely adopt the corresponding sulfur-plugging relief measures to ensure the normal production of gas wells.

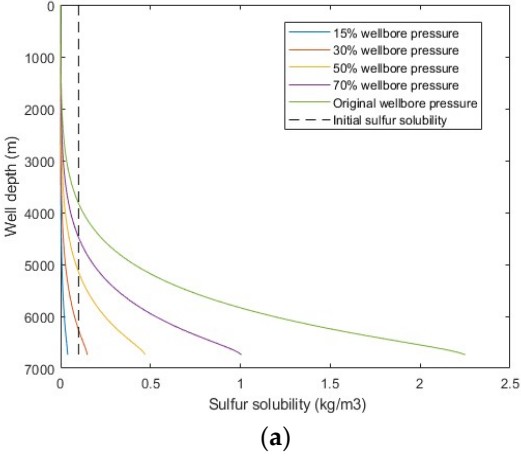
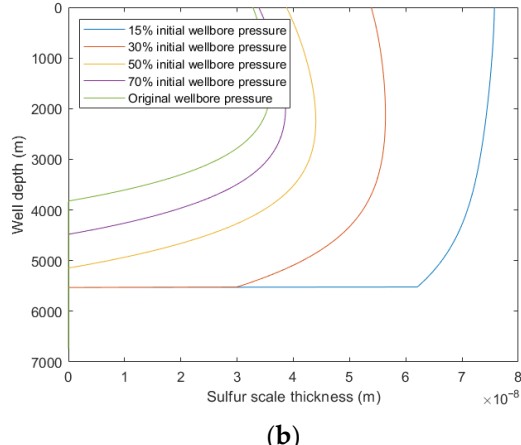

(**a**)

(**b**)

**Figure 11.** Calculated results with: (**a**) sulfur solubility distribution at different wellbore pressure; (**b**) sulfur scale thickness distribution at different wellbore pressure.

### 5.2.4. H$_2$S Content

In the middle and late stages of sour-gas well production, the H$_2$S content will continue to increase, which will change the density, viscosity, and sulfur solubility of the produced natural gas in the wellbore. The impact of H$_2$S content on sulfur deposition in wellbore shows that with the increases of H$_2$S content, the sulfur solubility increased, deposition rate decreased, and the sulfur deposition location is closer to the wellhead. It relieves the occurrence of sulfur deposition in the middle and late stages of gas well production to a certain extent. However, the increases of H$_2$S content will inevitably bring more serious corrosion to equipment and pipelines, and the natural gas purification process will also have a burden. It is necessary to pay attention to the possible harm and safety risks of these problems.

### 5.2.5. Well Trajectory

Three well trajectory types are simulated in this section: vertical well, horizontal well, and deviated well. The basic well trajectory parameters are shown in Table 7. And the simulation results are shown in Figure 12.

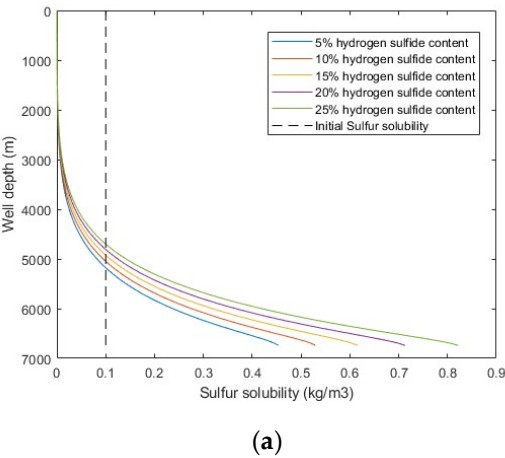
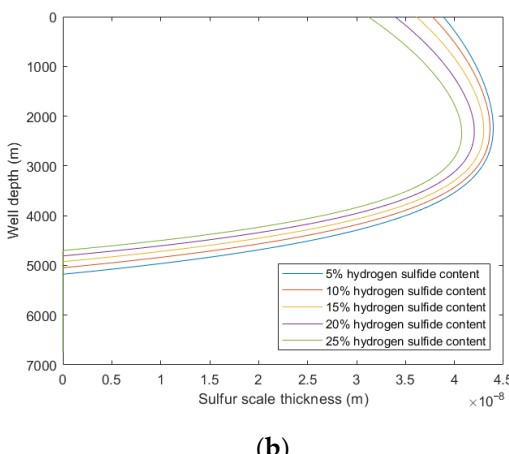

(**a**)

(**b**)

**Figure 12.** Calculated results with: (**a**) sulfur solubility distribution at different H$_2$S content; (**b**) sulfur scale thickness distribution at different H$_2$S content.

**Table 7.** Basic well trajectory parameters.

| Well Parameters | Vertical Well | Horizontal Well | Deviated Well |
| --- | --- | --- | --- |
| Well depth (m) | 7000 | 7000 | 7000 |
| Kickoff point (m) | / | 5000 | 5000 |
| Inclination angle (°) | 0 | 90 | 45 |

From Figure 13a,b, it was observed that the temperature comparison of each well trajectory type is horizontal well > deviated well > vertical well, the bottom hole pressure comparison is horizontal well < deviated well < vertical well, and the difference is mainly reflected in the build-up section. This is because with the increase of wellbore inclination angle, the actual vertical depth of the build-up section decreases, the heat conduction and emission of the wellbore to the formation decreases, and the pressure drop of fluid due to gravity and kinetic energy decreases. It can be seen from Figure 13c,d that with the increase of well inclination angle, the sulfur precipitation location is closer to the wellhead, and the sulfur scale thickness is smaller. Combined with Figure 13a,b, it is shown that temperature has a greater effect on sulfur deposition than pressure. Since the gas well productivity of horizontal wells and deviated wells is often greater than that of vertical wells, it is suggested that horizontal wells or highly deviated wells should be selected as the production well types. In addition, heating and insulation measures can be taken for the wellbore to reduce the occurrence of sulfur deposition.

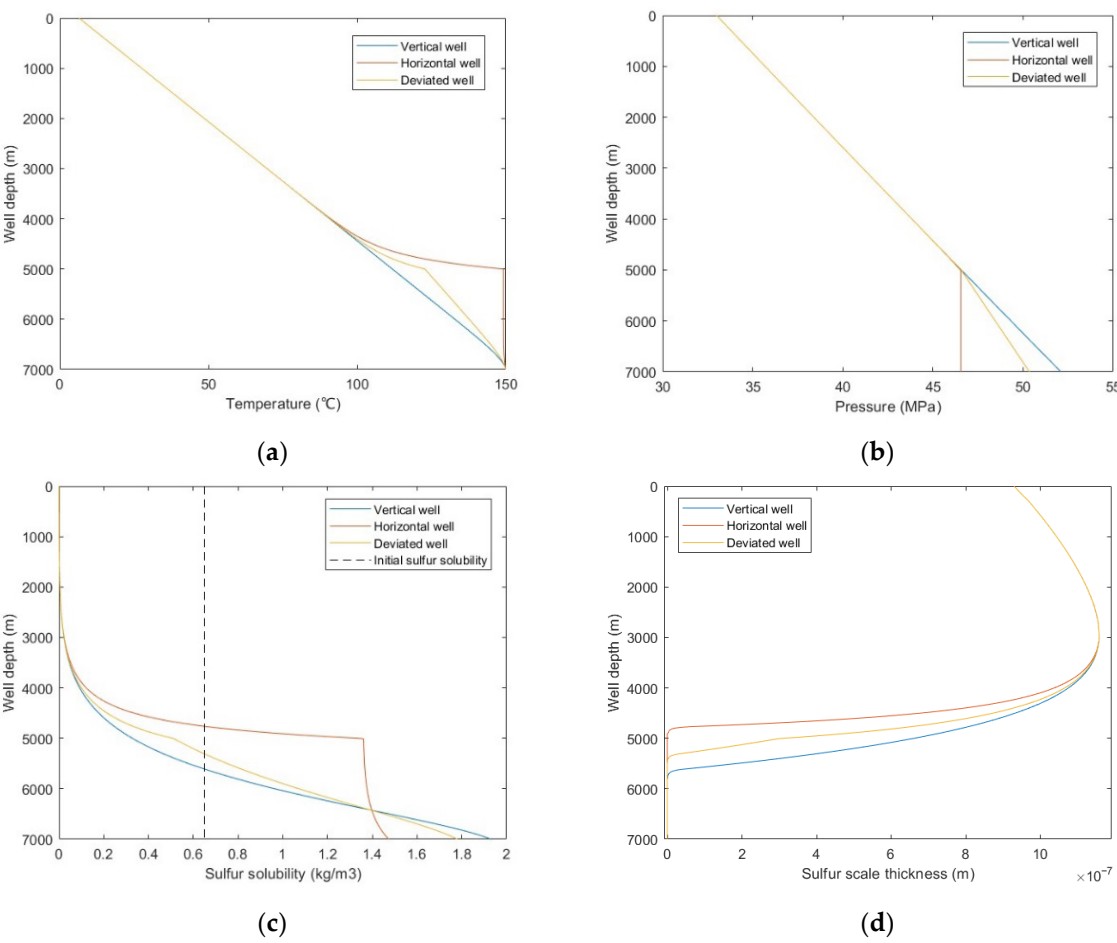

**Figure 13.** Calculated results with: (**a**) temperature distribution of different well trajectory types; (**b**) pressure distribution of different well trajectory types; (**c**) sulfur solubility distribution of different well trajectory types; (**d**) sulfur scale thickness distribution of different well trajectory types.

## 6. Conclusions

In this study, a numerical simulation procedure of sulfur deposition in the sour-gas well is proposed. The following conclusions are obtained:

(a) The wellbore sulfur deposition prediction model, pressure model, and transient temperature model considering the influence of different wellbore inclination angles were established. By coupling these models, the numerical simulation of sulfur deposition in the high-sulfur gas well was carried out. The average error between the calculated value and the measured value is 3.61%, indicating that the model has good calculation accuracy and engineering application value.

(b) The proposed methodology was used to simulate four sour-gas wells in China, and the results show that sulfur deposition are not formed in these wells, but elemental sulfur is precipitated. The influence of wellbore inclination angle on critical sulfur-carrying velocity is analyzed, and the results show that the critical sulfur-carrying velocity increases first and then decreases with the increase of inclination angle, and the maximum critical sulfur-carrying velocity is about 20% larger than that of the vertical well section.

(c) The effects of production time, gas production rate, wellbore pressure, $H_2S$ content, and well trajectory on wellbore sulfur deposition were analyzed. The results show that the wellbore sulfur deposition increases with the increase of production time and the decrease of wellbore pressure and decreases with the increase of gas production rate, $H_2S$ content, and inclination angle. Additionally, when gas production flow velocity exceeds the critical sulfur-carrying velocity, sulfur deposition will not occur in the wellbore.

(d) The technological measures to control sulfur deposition and prevent sulfur plugging of wellbore were presented. a. Adjust production systems dynamically to avoid gas wells under the same production system for too long. b. Use a pre-cleaning process where sulfur plugging may occur. c. Keep the gas production flow velocity greater than the critical sulfur-carrying velocity as far as possible. d. In the middle and late stages of gas well production, the production dynamics should be observed carefully and the technical measures to prevent sulfur deposition in wellbore should be implemented closely. e. Use the well type with large inclination angle for production. In addition, heat and insulation measures should be applied to the wellbore.

**Author Contributions:** Conceptualization, X.G.; methodology, X.G. and P.W.; validation, J.M. and C.J.; investigation, P.W.; resources, X.G.; data curation, P.W.; writing—original draft preparation, P.W.; writing—review and editing, X.G.; project administration, J.M. and C.J.; funding acquisition, X.G. All authors have read and agreed to the published version of the manuscript.

**Funding:** This research was funded by the General Program of National Natural Science Foundation of China, grant number 51874249, the State Key Program of National Natural Science Foundation of Sichuan Province, grant number 2022NSFSC0019, and the Joint Funds of National Natural Science Foundation of China, grant number U21A20105.

**Institutional Review Board Statement:** Not applicable.

**Informed Consent Statement:** Not applicable.

**Data Availability Statement:** The data presented in this study are available through email upon request to the corresponding author.

**Conflicts of Interest:** The authors declare no conflict of interest.

## Nomenclature

| | |
|---|---|
| $C$ | Sulfur solubility of gas, kg/m$^3$ |
| $C_o$ | Initial sulfur solubility, kg/m$^3$ |
| $C_s$ | Instantaneous sulfur solubility, kg/m$^3$ |
| $\rho_g$ | Natural gas density, kg/m$^3$ |

| | |
|---|---|
| $T$ | Temperature, K |
| $G$ | Gravity, N |
| $\rho_s$ | Sulfur density, kg/m$^3$ |
| $V_s$ | Sulfur volume, m$^3$ |
| g | Gravitational acceleration, m/s$^2$ |
| $F_f$ | Buoyancy, N |
| $\rho_m$ | Sour-gas density, kg/m$^3$ |
| $F_D$ | Drag force, N |
| $C_D$ | Drag coefficient |
| $S$ | Cross-sectional area of sulfur, m$^2$ |
| $v_m$ | Sour-gas flow velocity, m/s |
| $v_s$ | Sulfur flow velocity, m/s |
| $F_N$ | Support force, N |
| $\theta$ | Complementary Angle of wellbore inclination angle, ° |
| $f$ | Friction, N |
| $\lambda$ | Friction coefficient |
| $v_{gcr}$ | Critical sulfur-carrying velocity, m/s |
| $h_s$ | Height of the sulfur droplet, m |
| $\sigma$ | Interfacial tension of sulfur droplets, N/m |
| $N_{we}$ | Critical Weber number |
| $d_s$ | Diameter of sulfur particle, m |
| $Re$ | Reynolds number |
| $D$ | Wellbore diameter, m |
| $\mu_m$ | Sour-gas viscosity, Pa·s |
| $J_s$ | Transfer mass of sulfur, kg |
| $k_s$ | Mass transfer coefficient of sulfur, m/s |
| $M_s$ | Molecular weight of sulfur |
| $x_s$ | Mole content of sulfur in sour gas, mol/kg |
| $x_s{}^{jb}$ | Equilibrium molarity of sulfur, mol/kg |
| $j_D$ | Mass transfer factor |
| $S_C$ | Smith number of gas flow |
| $D_s{}^m$ | Diffusion coefficient of sulfur in sour gas, m$^2$/s |
| $p$ | Pressure, MPa |
| $z$ | Section length, m |
| $A$ | Wellbore circulation area, m$^2$ |
| $F_z$ | External forces, N |
| $Q_s$ | Amount of sulfur precipitated, kg/s |
| $Q_g$ | Gas production rate, m$^3$/s |
| $h$ | Enthalpy of sour gas, J/kg |
| $W$ | Mass flow rate of sour gas, kg/s |
| $q$ | Heat loss, J |
| $c_{pm}$ | Specific heat of sour gas, J/(kg·K) |
| $\alpha_H$ | Joule Thomson coefficient |
| $T_1(r,t)$ | Temperature function from wellbore to formation |
| $t$ | Production time, s |
| $r$ | Distance from wellbore center, m |
| $\alpha_1$ | Thermal conductivity of wellbore to formation, m$^2$/s |
| $U_{to}$ | Heat transfer coefficient of wellbore to formation, W/(m·K) |
| $(\rho c)_h$ | Heat capacity per unit volume of wellbore to formation, J/(m$^3$·K) |
| $r_h$ | Distance of cement sheath outer wall to wellbore center, m |
| $T_D$ | Dimensionless time function |
| $T_{ei}$ | Formation temperature, K |
| $t_D$ | Dimensionless time |
| $R_j{}^{-1}$ | Total wellbore heat transfer coefficient without sulfur scale, W/(m·K) |
| $r_{to}$ | Distance of tubing outer wall to wellbore center, m |
| $r_{ti}$ | Distance of tubing inner wall to wellbore center, m |
| $\Delta r$ | Sulfur scale thickness, m |
| $T_f$ | Wellbore temperature, K |
| $g_T$ | Geothermal gradient, K/m |
| $T_i$ | Bottomhole temperature, K |

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
