# Peer review of "Numerical Simulation of Sulfur Deposition in Wellbore of Sour-Gas Reservoir"

_processes, doi:10.3390/pr10091743_

Round 1

Reviewer 1 Report

The paper would be a nice contribution in Oil and Gas Reservoir Geology. From my point of view, I suggest the following points.

* The abstract is too short and incomplete. 

Introduction need to re-writing and please improve the important of the study

* Line 36 to correct : Abou-Kassem and Jamal [2] analyzed xxxxx

temperature is an influence factor of model. Please give the temperature distribution around wellbore. Especially, temperatures in wellbore and in the formation.

In the discussion, well trajectory should be discussed.

Additionally, the language need to be touched up.

The paper is very nice and I hope to see it published in the Processes Journal.

Good luck!

Reviewer 2 Report

In recent years, more and more attention in the world community has been paid to the issues of efficient production and further rational use of gas produced from wells. It should be noted that in the produced gas there are different concentrations of compounds of chemical elements, including sulfur compounds. To reduce the negative impact of sulfur compounds in gas production, it is necessary to develop an integrated methodology that involves mathematical modeling of sulfur deposits to increase the productivity of sour gas wells. The authors' article is devoted to an important and topical issue related to the development of an effective technique for mathematical modeling of sulfur deposition in a sour gas well. The technique has high accuracy, and the average error of the calculated results for it is 3.61%. An important aspect of the work, in my opinion, is that the modeling carried out by the authors has a number of significant advantages compared to analogues. The research presented in the paper is undoubtedly of interest to readers in the field under consideration.

However, it would be necessary to clarify a number of comments that are available to the article:

1. Section “2.3. Critical liquid sulfur carrying velocity" could be slightly reduced in terms of bringing a number of formulas, in particular, (4, 5, 6), giving a link to literary sources.

2. The article should have dwelled in more detail on the analysis of figures 3, 4, namely, to explain whether the presented diagrams are applicable for various section conditions?

3. It was necessary to provide a more detailed description of the block diagram (Figure 5).

4. Figure 6 should present predictive mathematical models of the parameters studied in the article.

5. Based on the results shown in Figures 10, 11, 12, it would be necessary to bring universal models for various conditions of sour gas wells

6. It is not entirely clear from the article on which gas wells the methodology developed by the authors was tested.

7. The list of references could include studies by Russian and American scientists, since the problems posed in the article are widely discussed in the Russian and American scientific community.

Round 2

Reviewer 1 Report

I would like to thank the authors for addressing all the comments/suggestions. I would like to inform you that I accept the paper for publication in Processes in its present form. Congratulations.